# Relations among Socially Prescribed Perfectionism, Career Stress, Mental Health, and Mindfulness in Korean College Students

**DOI:** 10.3390/ijerph182212248

**Published:** 2021-11-22

**Authors:** Sukkyung You, Jieun Yoo

**Affiliations:** 1College of Education, Hankuk University of Foreign Studies, 270 Imun-dong, Dongdaemun-gu, Seoul 130-791, Korea; skyou@hufs.ac.kr; 2Department of Christian Education, Anyang University, 22 Samdeok-ro, 37 Beon-gil, Manan-gu, Anyang 14028, Korea

**Keywords:** socially prescribed perfectionism, career stress, life satisfaction, depression, mindfulness

## Abstract

Korean young adults are exposed to higher career stress than ever before, and such stress exerts a negative impact on mental health outcomes. The present study aimed to understand the mediating effect of career stress on the relationship between socially prescribed perfectionism and mental health using a sample of 420 Korean college students. The present study also investigated the moderating role of mindfulness in the mediated pathways across gender groups. This study’s results showed that there are considerable gender differences in this relationship. Career stress significantly mediates the relationship between socially prescribed perfectionism and depression and life satisfaction only for females. Study findings also indicated that the moderating effect of mindfulness was more remarkable for female students than for male students. Implications and future directions are discussed.

## 1. Introduction

Recent data showed that over 20% of young Koreans aged 25 to 29 were unemployed, ranking highest among OECD countries in 2018 [1]. It has been reported that their career stress was one of the serious social issues in Korea [2]. Korean young adults were exposed to higher career stress than ever before, and such stress negatively impacted mental health outcomes [3].

Recently, extant research suggested the importance of emotional and personal trait factors to influence stress; in particular, it was suggested that people with a higher level of perfectionism are likely to experience a higher stress level [4]. Previous research supported that perfectionism is associated with career stress [3]. The moderating role of mindfulness has been receiving attention as a potential protective factor related to stress, depression, and life satisfaction [5,6]. However, few studies have investigated the effect of perfectionism on career stress and the effect of career stress on mental health, such as depression and life satisfaction. The potential mechanisms that might explain these relationships are still unclear; the mechanism underlying perfectionism, life satisfaction, and depression with career stress as a mediator and mindfulness as a moderator has not been examined thoroughly. Therefore, previous studies suggest understanding the relationship between perfectionism, career stress, mental health outcomes (e.g., life satisfaction, depression), and mindfulness in young adults.

### Perfectionism, Career Stress, Life Satisfaction, Depression, and Mindfulness

With the growing competition to get a job, Korean college students might be forced to excel and be perfect under Korean comparative social and economic situations [3]. Frost et al. [7] saw perfectionism as an indicator of competence for modern people living in today’s competitive society. In general, perfectionism is understood as a personality trait and has presented two broad aspects: the positive and the opposing standpoints. According to Hamachek [8], normal perfectionists are those individuals who hold high personal standards but accept the fact that these standards will not always be attained. In contrast, their neurotic counterparts are those individuals who report excessively high standards but are incapable of feelings of fulfillment, even when those standards are attained. Thus, maladaptive perfectionism is conceived to the negative subdimension of perfectionism. Moreover, defined as “people’s belief or perception that significant others have unrealistic standards for them, evaluate them stringently, and exert pressure on them to be perfect” [9]. 

Previous research supported that adaptive perfectionists reported significantly higher global life satisfaction than maladaptive perfectionists [10]. Seong and Hong [11] also found that socially prescribed perfectionism was negatively related to the subjective well-being of Korean undergraduate students. This finding would suggest that a person with a high level of maladaptive perfectionism would likely experience a decreased level of life satisfaction. Such a large body of literature has consistently studied the relationship between perfectionism and depressive symptoms. Several studies showed that people with maladaptive perfectionism experienced more distress than people with adaptive and non-perfectionism [10]. Maladaptive perfectionism was significantly associated with higher levels of depression among university students [3]. 

The association between maladaptive perfectionism and perceived stress has been well explored [10]. It was suggested that higher levels of maladaptive perfectionism were associated with higher levels of career stress of Korean college students [3]. Career stress refers to a condition related to employment after graduation, in which physical and psychological equilibrium is destroyed and replaced by feelings of crisis, tension, or anxiety [6]. Therefore, the career stress to seek a job was associated negatively with mental health and subjective well-being. As the perceived stress was well understood to be a predictor for low life satisfaction, it was found that career stress was negatively correlated with life satisfaction [12,13,14]. Some researchers reported that career indecision was related to depressive symptoms [3]. With a Korean college sample, career stress to seek a job was identified as a significant predictor of depression of college students [3]. The adverse effects of job-seeking stress on depression were decreased by resilience [13].

With its roots in ancient Buddhist meditation practices, mindfulness has been conceptualized as a form or state of present-moment attention or these qualities of awareness in terms of contemporary psychology [15]. Although individuals can reduce immediate emotional responses and respond appropriately to stressful situations by developing the ability to eliminate dysfunctional behavior patterns through mindfulness-based training [16], mindfulness might be explained as a dispositional trait or a trait-like general tendency [15]. A volume of studies supported that mindfulness was related to higher levels of life satisfaction and lower levels of stress and depression; mainly, mindfulness played the moderated role between career stress and depression [6]. This finding would suggest that people with a high level of mindfulness would tend to experience a different level of career stress, life satisfaction, and depression than those with a low level of mindfulness.

Despite the accumulated empirical findings of the association of perfectionism, career stress, mental health, and mindfulness in previous literature, more integrative links would be explored to understand the relationship among those related variables. Based on the extant research, we hypothesized that career stress might act as a mediator of the impact of maladaptive perfectionism on life satisfaction and depression. We hypothesized that mindfulness would moderate structural pathways among maladaptive perfectionism, career stress, depression, and life satisfaction. Furthermore, we would examine the difference between moderated mediational pathways by gender. 

## 2. Materials and Methods

### 2.1. Participants

A total of 420 university students were solicited to participate in this study in Seoul, Korea. The convenient sampling technique was utilized to collect the sample for the study. The sample was characterized by its gender ratio (64.3% female). The mean age of the sample was 21.63 years (SD = 2.61 years). 

### 2.2. Procedure

Instructors at the College of Social Sciences were contacted to obtain permission to recruit the study participants from their classes. Once approval was granted, trained research assistants distributed an informative letter describing the study. Those who consented to participate completed the survey in the paper-and-pencil format during class. All measures were provided in Korean and validated for Korean languages. Students were told not to write their names on the survey to protect their anonymity. The surveys took approximately 20–30 min to complete.

### 2.3. Measures 

#### 2.3.1. Career Stress

The Revised Life Stress Scale for College Students (RLSS-CS) was used to measure career stress [17]. This scale contains seven items (e.g., I am nervous because of unemployment problems these days). The scale was rated using a 5-point Likert scale (1 = never to 5 = very often). For this sample, Cronbach’s alpha was 0.80. Other studies of Korean college students also found significant associations between this measure and suicidal ideation [18], providing evidence for the instrument’s validity for use with this population.

#### 2.3.2. Socially Prescribed Perfectionism

The subscale of the Multidimensional Perfectionism Scales (HMPS) [9] was used to measure the level of Socially Prescribed Perfectionism. This scale contains 20 items (e.g., The better I do, the more is expected from me). The scale was rated using a 5-point Likert scale (1 = strongly disagree to 5 = strongly agree). For this sample, Cronbach’s alpha was 0.83. Other studies with Korean college students also found significant associations with maladaptive perfectionism and stress [19], providing evidence for the instrument’s validity for use with this population.

#### 2.3.3. Depression

Depression was measured using The Center for Epidemiology Studies-Depression (CES-D) [20]. This scale contains 20 items (e.g., I find things that used to be fine with me annoying and bothersome). The scale was rated using a 4-point Likert scale (0 = Not at all to 3 = 5–7 times). A high score on the scale implies a high level of depression. For this sample, Cronbach’s alpha was 0.81, which compares to 0.91–0.92 in previous studies [21,22]. Other research findings have shown significant associations between this measure and related variables, such as anxiety and negative affect [23].

#### 2.3.4. Life Satisfaction

Life satisfaction was measured using the Satisfaction with Life Scale (SWLS) [24]. This scale contains five items (e.g., I am satisfied with my life). The scale was rated using a 7-point Likert scale (1 = strongly disagree to 7 = strongly agree). For this sample, Cronbach’s alpha was 0.81. Other studies of Korean adults also found significant associations with life satisfaction and job stress [14], providing evidence for the instrument’s validity for use with this population. 

#### 2.3.5. Mindfulness

Mindfulness was measured using The Mindful Attention Awareness Scale [25]. This scale contains 15 items (e.g., I find it difficult to stay focused on what is happening in the present). The scale was rated using a 6-point Likert scale (1 = strongly disagree to 6 = *strongly agree*). For this sample, Cronbach’s alpha was 0.80. Other studies of Korean adults also found significant associations with depression and job stress [6], providing evidence for the instrument’s validity for use with this population. 

### 2.4. Analyses

Structural Equation Modeling (SEM) was used to assess the hypothesized structural relationships between the latent variables. SEM was selected because it is an appropriate analytical approach specifying directionality for the variables of interest while also generating the flexibility to test causal relationships. Two mediational models were tested to compare and derive the best one. The model’s fit was assessed based on several criteria: the Comparative Fit Index (CFI) [26], the non-normed fit index (NNFI) [27], and the root means square error of approximation (RMSEA) [28]. Values lower than 0.06 for the RMSEA and close to 0.95 for the NNFI and CFI were used to determine a good-fit model [29].

To test the significance of the mediating effects, we used the bootstrapping method outlined by Shrout and Bolger [30]. This method utilizes repeated sampling from the data set and estimating the indirect effect in each resampled data set. This approach is recommended over the traditional Sobel test, since it makes no assumptions regarding the shape of the sampling distribution of the indirect effect [31]. All analyses were conducted using AMOS [32].

## 3. Results

### 3.1. Descriptive Statistics

The variables’ correlations, mean, and standard deviation are provided in Table 1. According to the guidelines of severe non-normality (i.e., skewness > 3; kurtosis > 10) proposed by Curran, West, and Finch [33], the normality assumption of all the variables were well met, where the skewness values were less than 3 and kurtosis values were less than 10.

### 3.2. Testing the Mediational Models

For the male group, the partial mediational model yielded an overall χ^2^(72) value of 94.03, with CFI = 0.954, NNFI = 0.950, and RMSEA = 0.064 and the full mediational model yielding an overall χ^2^(74) value of 100.78, with CFI = 0.944, NNFI = 0.940, and RMSEA = 0.079. For the female group, the partial mediational model yielded an overall χ^2^(72) value of 132.25, with CFI = 0.950, NNFI = 0.954, and RMSEA = 0.060 and the full mediational model yielding an overall χ^2^(74) value of 132.25, with CFI = 0.929, NNFI = 0.936, and RMSEA = 0.070. A chi-square difference test was conducted that supported the partial meditational model for both groups. The fit of the final model was deemed acceptable in terms of three fit indices. The standardized parameter estimates for this model are presented in Figure 1. 

These results indicate that maladaptive perfectionism has significant direct effects on depression among male undergraduate students. However, maladaptive perfectionism has significant direct effects on life satisfaction for females. Regarding the mediating effects, the bootstrap test results vary across the level of mindfulness for females. Specifically, the effect of maladaptive perfectionism on depression via career stress for females with higher levels of mindfulness was significant (*β* = 0.11, *p* < 0.05). Meanwhile, maladaptive perfectionism on life satisfaction and depression for females with lower levels of mindfulness (*β* = 0.08, and *β* = 0.09, *p* < 0.05, respectively) via career stress were all significant. 

## 4. Discussion 

The present study aimed to understand the mediating effect of career stress on the relationship among maladaptive perfectionism, life satisfaction, and depression using a sample of Korean college students. The present study also investigated the moderating role of mindfulness in the mediated pathways across gender groups. 

In this sample, female college students seemed to have higher average scores of maladaptive perfectionism, depression, and mindfulness, and lower average scores of career stress and life satisfaction compared to male students. This finding aligned with several other findings showing that female students showed more depression among Korean university students than males [3]. The correlations findings showed, as expected, that maladaptive perfectionism was positively related to career stress and depression. In contrast, it was negatively related to life satisfaction and mindfulness for both groups, which is consistent with the previous studies [18].

The SEM results indicated that mindfulness had a moderating effect, supporting previous studies [6]. The higher the mindfulness, the weaker the associations among the variables. This means that mindfulness had a significant influence on relationships among maladaptive perfectionism, career stress, and mental health. These findings also showed that the paths for the female group appeared much more clearly. This is consistent with a previous study in that the effect of mindfulness was higher for female students than for male students [34].

Results also showed that career stress did not significantly mediate the relationship between maladaptive perfectionism and mental health for males regardless of mindfulness. In addition, it is notable that career stress only mediated the relationship between maladaptive perfectionism and depression for females with higher levels of mindfulness, but it mediated the relationship between maladaptive perfectionism and depression and life satisfaction for females with lower levels of mindfulness. Results also suggested that maladaptive perfectionism has directly and indirectly affected depression and life satisfaction for the female group with a lower level of mindfulness. Our result partially supported the findings in the previous studies [6].

One possible explanation for current results was that maladaptive perfectionism involves excessively high standards. However, unfulfillment allows one to become more hostile and discouraged. Therefore, people with a high level of maladaptive perfectionism tend to struggle against what they cannot fulfill their goals, such as job seeking [35]. The emphasis on job preparation for female college students in today’s Korean society has strengthened the belief in perfectionism to advance oneself or others. Amidst poor Korean employment conditions of unequal ratios across gender groups, females must meet higher standards than males, so they were forced to pursue perfectionism themselves [36]. With such demand, female college students have no choice but to pursue this perfectionism themselves. Their career stress to seek a job is deteriorating, and their depression and life satisfaction are worsening. That is, their focus on a socially given adverse evaluation gives them more psychological distress, because of which it is not easy to fully express their abilities.

This result confirmed that mindfulness intervention was expected to be effective in alleviating maladaptive perfectionism, which attempted to be perfect in response to the expectations of others, because it paid attention to thoughts and emotions with the ability to pay attention to the present moment. Mindfulness is the critical observation of an accident and accepting that automatic thoughts and feelings are merely reflections, not actual reflections [15]. Therefore, to reduce the career stress related to depression and life satisfaction, it is necessary to seek active counseling interventions to lower socially prescribed perfectionism and higher mindfulness. Furthermore, it might help students in distress if teachers and parents also share their failure experiences and coping strategies to tackle such situations.

There were several following limitations to be interpreted carefully. This study was designed as a cross-sectional study of relationships among variables at a one time point. Given the sample composition, the characteristics of college students who participated in this study limited the generalizability of the research findings. Finally, the present findings were based on participants’ self-report surveys. Despite these limitations, the present study extended the research related to mental health outcomes with college-aged students. The empirical study dealing with maladaptive perfectionism, career stress, depression, life satisfaction, and mindfulness has been very scarce. Thus, it is significant because it has prepared primary data for follow-up studies by verifying the relationship among those variables for college students.

## 5. Conclusions

To sum up, the present study substantially provides our insight into a complicated interplay among mindfulness and mental health on an unrepresented study sample. The current study showed that career stress significantly mediates the relationship between socially prescribed perfectionism and depression and life satisfaction only for females. Study findings also indicated that the moderating effect of mindfulness was more remarkable for female students than for male students. The findings regarding gender differences indicate a significant difference in relations between mindfulness and mental health between males and females, suggesting that paying attention in the present moment is an essential source of life satisfaction and depression, especially for females. The current results suggested several counseling implications that can inform preventative measures and therapeutic interventions targeting Korean female college students. College counselors should understand the role of maladaptive perfectionism to encourage female college students to pursue positive and flexible thoughts that permit them to be imperfect.

## Figures and Tables

**Figure 1 ijerph-18-12248-f001:**
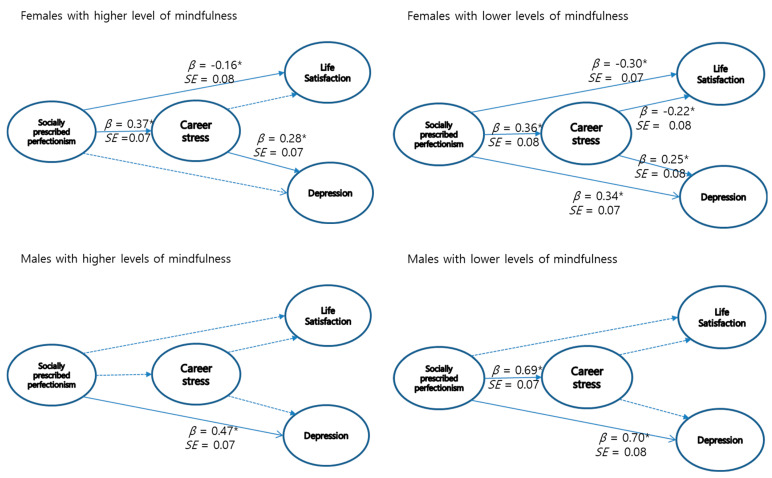
Final model estimation with standardized beta weight (*SE*). Note. * *p* < 0.05; significant path coefficients are shown in bold line; dotted line = non-significant.

**Table 1 ijerph-18-12248-t001:** Correlations and descriptive statistics for study variables.

	1	2	3	4	5
1. Socially prescribed perfectionism ^α^	1	0.42 *	−0.42 *	0.44 *	−0.57 *
2. Career stress ^α^	0.49 *	1	−0.29 *	0.37 *	−0.25 *
3. Life satisfaction ^α^	−0.14 *	−0.22 *	1	−0.43 *	0.34 *
4. Depression ^α^	0.58 *	0.51 *	−0.16 *	1	−0.45
5. Mindfulness ^α^	−0.25 *	−0.17 *	0.51 *	−0.30 *	1
*Means* (*SD*)					
Male	2.69(0.60)	2.14(0.62)	3.38(0.74)	1.84(0.58)	2.72(0.84)
Female	2.89(0.78)	1.95(0.60)	3.19(0.82)	1.89(0.67)	2.95(0.79)
*Cronbach’s alphas*					
Male	0.79	0.78	0.78	0.79	0.81
Female	0.82	0.81	0.83	0.83	0.79

Note. Correlations for females are above diagonal; ^α^ Gender difference is significant at * *p* < 0.05.

## Data Availability

Data are available upon request from the corresponding author. The data are not publicly available due to privacy or ethical restrictions.

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
