# Peer review of "Relations among Socially Prescribed Perfectionism, Career Stress, Mental Health, and Mindfulness in Korean College Students"

_ijerph, 2021, doi:10.3390/ijerph182212248_

Round 1

Reviewer 1 Report

The study is interesting and deserves to be published, but needs the following mainly methodological improvements:
The description of the type of sampling and the reliability of the study.
Explanation of the procedure.
The tables with the weight of the values of the items and in the SEM of the latent and observed variables.
More explanation of the results and discussion of the results obtained.

Author Response

The description of the type of sampling and the reliability of the study.
Explanation of the procedure.-->As suggested, we have added the information mentioned above.
The tables with the weight of the values of the items and in the SEM of the latent and observed variables.-->As suggested, we have added the information mentioned above.
More explanation of the results and discussion of the results obtained.-->As suggested, we have added more information regarding results and discussion of the results.

Reviewer 2 Report

Dear Authors,

This was a very interesting read that adresses very important issue of stress among young Korean population, and this isn't a secret that the issue is there and needs to be faced. The methodology used is suitable, and the results (in general) support the conclusions. However, few changes have to be made before the article is suitable for publication in IJERPH.

  1. English changes are required. The style here and there is not very good and could be improved by an English collegue or online language improving tools.
  2. Were the questionnaires used validated for Korean language? If yes, please state so in the methodology section. If no, please state so in the study limitations section.
  3. Please remember that this is INTERNATIONAL Journal of Environmental Research and Public Health. Please check the manuscript if you have explained to the international audience what do certain Korean stuff mean, for example - what is Statistics Korea? Is this a national office, or some kind of NGO? Everything needs to be clear for international viewers.
  4. I think that citing articles in conlusion is a flaw that needs to be fixed. Please revise the conlusions section so it contains only what you directly conclude from the results presented.

Good luck revising the article. I think it will fit very well to IJERPH after the comments above will get addressed. 

Best Regards,

Reviewer

Author Response

  1. English changes are required. The style here and there is not very good and could be improved by an English collegue or online language improving tools. --> As suggested, we have had the paper proofread and polished.
  2. Were the questionnaires used validated for Korean language? If yes, please state so in the methodology section. If no, please state so in the study limitations section. -->Yes, the questionnaires used were validated in Korean language. As suggested, we have added the information mentioned above.
  3. Please remember that this is INTERNATIONAL Journal of Environmental Research and Public Health. Please check the manuscript if you have explained to the international audience what do certain Korean stuff mean, for example - what is Statistics Korea? Is this a national office, or some kind of NGO? Everything needs to be clear for international viewers. -->As suggested, we have revised sentences for international readers.
  4. I think that citing articles in conlusion is a flaw that needs to be fixed. Please revise the conlusions section so it contains only what you directly conclude from the results presented. -->As suggested, we have deleted citations and revised the conclusion section.